# On Translation and Reconstruction Guarantees of the Cycle-Consistent Generative Adversarial Networks

**Anish Chakrabarty**
Statistics and Mathematics Unit
Indian Statistical Institute, Kolkata
West Bengal, India

**Swagatam Das**[*]
Electronics and Communication Sciences Unit
Indian Statistical Institute, Kolkata
West Bengal, India

## Abstract

The task of unpaired image-to-image translation has witnessed a revolution with the introduction of the cycle-consistency loss to Generative Adversarial Networks (GANs). Numerous variants, with Cycle-Consistent Adversarial Network (CycleGAN) at their forefront, have shown remarkable empirical performance. The involvement of two unalike data spaces and the existence of multiple solution maps between them are some of the facets that make such architectures unique. In this study, we investigate the statistical properties of such unpaired data translator networks between distinct spaces, bearing the additional responsibility of cycle-consistency. In a density estimation setup, we derive sharp non-asymptotic bounds on the translation errors under suitably characterized models. This, in turn, points out sufficient regularity conditions that maps must obey to carry out successful translations. We further show that cycle-consistency is achieved as a consequence of the data being successfully generated in each space based on observations from the other. In a first-of-its-kind attempt, we also provide deterministic bounds on the cumulative reconstruction error. In the process, we establish tolerable upper bounds on the discrepancy responsible for ill-posedness in such networks.

## 1 Introduction

The overwhelming number of variants GANs [1] have inspired while catering to its vast application domains is a testament to its versatility. One such family of progenies having remarkable accolades of its own owes its genesis to the cycle-consistency constraint. Possibly the most influential one belonging to this group is CycleGAN [2]. It offers an unsupervised image-to-image (I2I) translation framework for unpaired observations hailing from unrelated data spaces. In terms of the architecture, both DualGAN [3] and DiscoGAN [4] are immediate relatives to CycleGAN. In the chassis of such networks lie two concurrent adversarial generation processes, commonly termed *translations*, regularized by a cyclic loss. This penalization ensures the reconstruction of input data from either space post-translation. In addition, models such as DTN [5] and UNIT [6] assume the existence of a shared latent space between the domains. This allows the restructuring of the model without altering the objective. By stacking multiple translator networks, SCAN [7] promises significant performance improvement, especially for high-resolution images. Some members of the family ([2], U-GAT-IT[8], [9]) also deploy an additional identity loss to remove tilt-shift in generated images. We call this broad class of I2I translation machines 'cycle-consistent networks'. The constraint of cycle-consistency should be primarily credited for the masterly generative capability of such models, from which tasks like style transfer, object transfiguration [2], and data augmentation [10] benefit immensely.

In this study, we intend to rise above the empirical evidences by providing a statistical backbone to the fact that cycle-consistent networks can simultaneously translate data both ways without losing the

---

[*]Corresponding Author (Email: `swagatam.das@isical.ac.in`)

36th Conference on Neural Information Processing Systems (NeurIPS 2022).

capacity to reconstruct. We call the two maps, operating in opposite directions, *Translators*. Both underlying distributions portraying purposeful image data, in the absence of conventional latent laws, call for such transformations to differ from usual generators used in vanilla GANs. Unlike GANs, there may exist non-unique solution maps bringing about 'zero' realized loss in this case [9]. As such, searching for translators that minimize the error is not sufficient. This fact motivates us to study the desirable regularities of the maps that facilitate the statistical convergence of output measures, which in turn define our notion of *consistency*. Theoretically, the concept of 'cycle-consistency' is analogous to 'regeneration' [11] in case of Variational Autoencoders [12]. In such inverse problems, maps reconstructing the input signal become sensitive to slight perturbations due to noise. A noisy output from earlier translations contributes to this ambiguity in the inverse generation process, formally known as 'ill-posedness' [13]. Theoretical insights regarding the source and admissible error margins of ill-posedness remain absent to date. Confronted with such challenges, this paper provides a fresh perspective on the theoretical machinery of the cycle-consistent adversarial networks. Our contributions can be summarized in the following way:

- We show that translators, based on deep ReLU networks, prevent the information provided by input empirical laws from dissipating during generation cycles [Proposition (1) and Remark (3)].

- In Theorem (1) and Corollary (1), we prove that the same translators not only achieve zero generation loss asymptotically but the generated sequence of distributions also converge to the target density almost surely.

- Under Sobolev-smooth input laws, we establish that the uses of $L^1$ norm and 1-Wasserstein distance in the cyclic loss are equivalent, attesting to Zhu *et al.*'s [2] observation that the latter does not improve performance [Theorem (3)].

- Furthermore, we prove that a network deploying the aforementioned translators achieves cycle-consistency as a consequence of translation consistency in both directions. [Theorem (3), (5)]

## 2   Related work

Playing catch-up to earlier empirical success, theoretical scrutiny of GANs fostered a series of notable works in recent years. Liu *et al.* [14] characterized the objective functions of several GAN architectures (f-GAN [15], WGAN [16], etc.) as *adversarial divergences*. This allowed them to analyze the convergence of generated distributions towards the target law in a unified framework. Meanwhile, Arora *et al.* [17] explored the expressiveness of generator networks and the generalization performance of GANs under the *neural net distance*. In a later work, however, we observed Zhang *et al.* [18] show improved results over both. Convergence and related asymptotic properties of the density estimates in a GAN setup can also be found in the parametric approach of [19]. On the other hand, error decomposition of the GAN-objective under both parametric and non-parametric regimes may lead to non-asymptotic concentration bounds. Several works followed this approach with various smoothness assumptions on the data distributions and the transformations involved [20, 21, 22]. A recent study of the same nature also focused on learning from low-dimensional latent laws using smooth maps [23]. One may also come across several GAN variants inspiring similar pursuits. Biau *et al.* [24] presented a comprehensive study of the convergence and related asymptotic properties of the parametric density estimates in a WGAN setup. From a non-parametric viewpoint, Haas *et al.* [25] derived deterministic upper bounds on the expected WGAN loss, under both conditional and unconditional generation processes. Lately, a non-asymptotic approach of a similar spirit has been utilized to establish risk bounds on the realized Bidirectional-GAN (BiGAN) error [26].

Cycle-consistent networks, despite marking a triumph in deep generative modeling, have not received such independent attention yet. This scarcity makes the existing attempts even more meaningful. Moriakov *et al.* [9] proved that multiple solutions to the CycleGAN problem exist, as a consequence of the existence of nontrivial automorphisms in either data space. Tiao *et al.* [27] pointed out that the cycle-consistency loss boils down to an expected posterior log-likelihood, in a Bayesian setup. On the contrary, the CycleGAN objective can also be recognized as the Unbalanced Gromov-Monge Divergence (UGMD), when the transformations are assumed to be isometric [28]. However, all the above studies refrain from exploring the statistical guarantees a cycle-consistent translator aims to provide by following its concurrent objectives. Our current work is a humble attempt to fill this gap.

# 3 Preliminaries

## 3.1 Notations

The two data spaces involved $\mathcal{X}$ and $\mathcal{Y}$, equipped with respective distances $c$ and $c'$, are considered to be Polish (i.e., separable and completely metrizable). A simple characterization of the same might be $\mathbb{R}^d$ for $d \geq 1$. Let $\mathscr{P}(\mathcal{X})$ denote the space of probability measures defined on $\mathcal{X}$. We refer to the set of measurable functions mapping $\mathcal{X}$ to $\mathcal{Y}$ as $\mathscr{F}(\mathcal{X}, \mathcal{Y})$. The 'forward' and 'backward' translator maps between the spaces are denoted by $F$ and $G$ respectively. Observe that, a probabilistic forward translator belongs to $\mathscr{F}(\mathcal{X}, \mathscr{P}(\mathcal{Y}))$. Similarly, $G \in \mathscr{F}(\mathcal{Y}, \mathscr{P}(\mathcal{X}))$. The two discriminator networks at both ends induce functions $D_X$ and $D_Y$, which play the role of critics in the two simultaneous adversarial games. For non-negative real sequences $\{a_n\}_{n \in \mathbb{N}}$ and $\{b_n\}_{n \in \mathbb{N}}$, the notation $a_n \precsim b_n$, or equivalently $a_n = \mathcal{O}(b_n)$, means that there exists a constant $C > 0$, such that $\limsup_{n \to \infty} \frac{a_n}{b_n} \leq C$. The *total variation distance* between measures $P, Q$ is represented by $\|P - Q\|_{TV}$. We also denote $\max\{x, y\}$ as $x \vee y$. Let us now introduce some concepts that we frequent in the upcoming discussion.

**Definition 1** (Pushforward Measure). *Let $\mu \in \mathscr{P}(\mathcal{X})$. For a measurable map $f : \mathcal{X} \to \mathcal{Y}$, we denote the pushforward measure of $\mu$ by $f_{\#}\mu \in \mathscr{P}(\mathcal{Y})$, defined as $f_{\#}\mu(\omega) = \mu(f^{-1}(\omega))$, where $\omega$ is a measurable set $\subset \mathcal{Y}$.*

**Definition 2** (Wasserstein Distance). *For a metric $c : \mathcal{X} \times \mathcal{X} \to \mathbb{R}_{\geq 0}$ and measures $P, Q \in \mathscr{P}(\mathcal{X})$, the $r^{th}$ Wasserstein Distance between $P$ and $Q$ is defined as*

$$W_c^r(P, Q) = \inf_{\gamma \in \Gamma(P,Q)} \left\{ \int_{\mathcal{X} \times \mathcal{X}} [c(x,y)]^r d\gamma(x,y) \right\}^{\frac{1}{r}},$$

*where $\Gamma(P, Q) = \left\{ \gamma \in \mathscr{P}(\mathcal{X} \times \mathcal{X}) : \int_{\mathcal{X}} \gamma(x,y) dy = P, \int_{\mathcal{X}} \gamma(x,y) dx = Q \right\}$ is the set of all measure couples between $P$ and $Q$; $r \in [1, \infty)$.*

**Remark 1.** In our analysis, we make extensive use of a particular case of this discrepancy measure, namely when $r = 1$. We also reiterate the fact that $W_c^1$ can be written as $W_c^1(P, Q) = \sup_{l \in \mathscr{L}_c^1} \left\{ \int_{\mathcal{X}} l(x) dP(x) - \int_{\mathcal{X}} l(x) dQ(x) \right\}$, where $\mathscr{L}_c^1 :=$ class of 1-Lipschitz functions with respect to $c$ [Remark 6.5 in [29]]. However, we adopt the notation $d_{\mathscr{L}_c^1}$ instead to maintain consistency with the other Integral Probability Metrics (IPMs).

## 3.2 Problem setup

Throughout our discussion, we denote the distributions at both ends by $\mu \in \mathscr{P}(\mathcal{X})$ and $\nu \in \mathscr{P}(\mathcal{Y})$ respectively. The adversarial loss that the backward generation process ($\mu \xleftarrow{G} \nu$) tries to minimize is given by,

$$\mathcal{L}_{D_X}(\mu, \nu, G) = \mathbb{E}_{x \sim \mu}[D_X(x)] - \mathbb{E}_{y \sim \nu}[D_X(G(y))].$$

The same convention leads to the forward generation ($\mu \xrightarrow{F} \nu$) loss, $\mathcal{L}_{D_Y}(\nu, \mu, F)$. The string tying these two processes together comes in the form of the cyclic loss. Based on our notations, it can be written as

$$\mathcal{L}_{cyc}(\mu, \nu, F, G) = \mathbb{E}_{x \sim \mu}\left[\left\|x - G(F(x))\right\|_1\right] + \mathbb{E}_{y \sim \nu}\left[\left\|y - F(G(y))\right\|_1\right],$$

where $\|\cdot\|_1$ represents the $L^1$ norm. We point out that this specific choice of the norm is based on the recommendation of Zhu *et al.* [2]. According to them, the usage of an adversarial loss instead does not improve regenerated image quality. Through the following illustration (Figure 1), we offer the reader a glimpse of our vision of concurrent translations and reconstructions.

A typical CycleGAN [2], or equivalently DiscoGAN [4] formulation, carries out the following optimization task:

$$\inf_{\substack{F \in \mathscr{F}(\mathcal{X}, \mathscr{P}(\mathcal{Y})) \\ G \in \mathscr{F}(\mathcal{Y}, \mathscr{P}(\mathcal{X}))}} \sup_{\substack{D_X \in \mathscr{L}_X \\ D_Y \in \mathscr{L}_Y}} \left\{ \mathcal{L}_{cyc}(\mu, \nu, F, G) + \lambda_1 \mathcal{L}_{D_X}(\mu, \nu, G) + \lambda_2 \mathcal{L}_{D_Y}(\nu, \mu, F) \right\}, \tag{1}$$

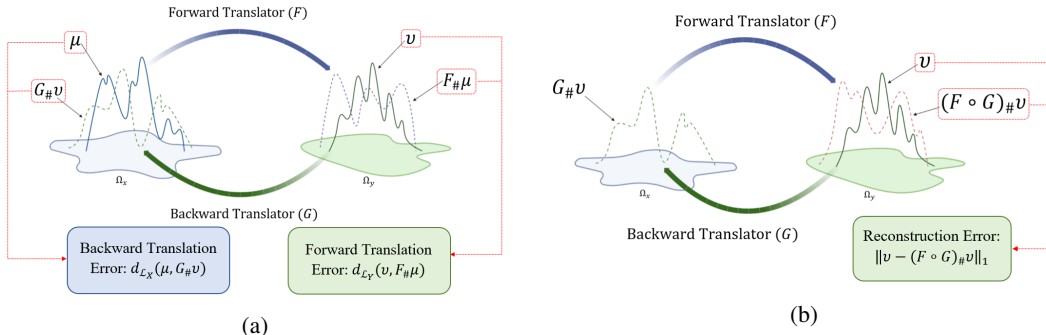

Figure 1: (a) Forward and backward translations with corresponding errors, (b) Reconstruction in the space $\mathcal{Y}$, all viewed through the glass of density estimation.

where $\mathscr{L}_X$ and $\mathscr{L}_Y$ are classes of discriminator functions and the maps $F$, $G$ are sculpted using translator networks. Also, $\lambda_1, \lambda_2 > 0$. Observe that, (1) can be rewritten as

$$\inf_{\substack{F \in \mathscr{F}(\mathcal{X}, \mathscr{P}(\mathcal{Y})) \\ G \in \mathscr{F}(\mathcal{Y}, \mathscr{P}(\mathcal{X}))}} \left\{ \mathcal{L}_{cyc}(\mu, \nu, F, G) + \lambda_1 \sup_{D_X \in \mathscr{L}_X} \mathcal{L}_{D_X}(\mu, \nu, G) + \lambda_2 \sup_{D_Y \in \mathscr{L}_Y} \mathcal{L}_{D_Y}(\nu, \mu, F) \right\}$$

$$\equiv \inf_{\substack{F \in \mathscr{F}(\mathcal{X}, \mathscr{P}(\mathcal{Y})) \\ G \in \mathscr{F}(\mathcal{Y}, \mathscr{P}(\mathcal{X}))}} \left\{ \mathcal{L}_{cyc}(\mu, \nu, F, G) + \lambda_1 d_{\mathscr{L}_X}(\mu, G_{\#}\nu) + \lambda_2 d_{\mathscr{L}_Y}(\nu, F_{\#}\mu) \right\}, \tag{2}$$

given that $d_{\mathcal{F}}(P, Q) = \sup_{f \in \mathcal{F}} \left\{ \mathbb{E}_P[f] - \mathbb{E}_Q[f] \right\}$. The only feature that differentiates the DualGAN [3] objective from (2) is the employment of the conventional generation technique based on noises. Furthermore, in an attempt to preserve the colour composition in images, models such as the extended CycleGAN [2, 9], U-GAT-IT [8] impose a constraint that penalizes the translators' tendency to move away from the identity. It is given by,

$$\mathcal{L}_{id}(\mu, \nu, F, G) = \mathbb{E}_{x \sim \mu} \left[ \left\| x - F(x) \right\|_1 \right] + \mathbb{E}_{y \sim \nu} \left[ \left\| y - G(y) \right\|_1 \right].$$

Observe that, such regularization is feasible only when the two data distributions are equi-dimensional. The map $F$ is built with the fundamental motivation of transforming $\mu$ into $\nu$. As such, the discrepancy $\left\| \mu - F_{\#}\mu \right\|_1$ should not be minimized beyond the difference between $\mu$ and $\nu$. We provide a detailed discussion on the same in the supplementary material.

## 4 Theoretical analysis

### 4.1 Data distributions

Depicting real images as observations from probability distributions is a completely theoretical construct. The representation provides practitioners with a refined view of the problem. Moreover, the transformed objective of density estimation has its benefits. Perhaps this is the idea that inspired the genesis of 'Roundtrip', a CycleGAN progeny [30]. In our study, we consider $\mathcal{X} \equiv \mathbb{R}^d$ and $\mathcal{Y} \equiv \mathbb{R}^k$; $d, k \in \mathbb{N}^+$. The two dimensions need not be equal in general. The consequences of the special case of equality will be discussed at a later stage. Let us now introduce some particular classes of functions that determine the nature of the data distributions under consideration.

We denote by $L_p(\mathbb{R}^d)$ the space of $p$-fold Lebesgue-integrable functions, equipped with the norm $\| \cdot \|_p$, $p \in [1, \infty)$. The notation for the space of uniformly continuous functions is $C_u(\mathbb{R}^d)$. Also, $\| \cdot \|_\infty$ stands for the uniform norm.

**Definition 3** (Sobolev Space [31]). *Let $\alpha = (\alpha_1, \alpha_2, ..., \alpha_d)$, $\alpha_i \in \mathbb{N}^+ \cup \{0\}$ such that $|\alpha| = \sum_{i=1}^{d} \alpha_i$. For $x \in \mathbb{R}^d$, the mixed partial weak differential operator of order $|\alpha|$ is given by $D^\alpha = \frac{\partial^{|\alpha|}}{\partial x_1^{\alpha_1} ... \partial x_d^{\alpha_d}}$. Based on these notations, for $p \in [1, \infty)$ and radius $L \in \mathbb{R}_{\geq 0}$*

$$\mathcal{W}_L^{m,p}(\mathbb{R}^d) = \left\{ f \in L_p(\mathbb{R}^d) : D^\alpha f \in L_p(\mathbb{R}^d) \forall |\alpha| \leq m : \|f\|_{\mathcal{W}^{m,p}} \equiv \|f\|_p + \sum_{|\alpha|=m} \|D^\alpha f\|_p < L \right\}$$

*defines the $L^p$-Sobolev Space of order $m$.*

**Remark 2.** Here, we also mention an extended class of functions, mainly when $p = \infty$. If $f : \mathbb{R}^d \to \mathbb{R}$ is differentiable at $x$, we write $D^\alpha f = f^{(\alpha)}$. We define

$$\mathcal{W}_L^{m,\infty}(\mathbb{R}^d) = \left\{ f \in C_u(\mathbb{R}^d) : f^{(\alpha)} \in C_u(\mathbb{R}^d) \forall |\alpha| \leq m : \|f\|_{\mathcal{W}^m} \equiv \|f\|_\infty + \sum_{|\alpha|=m} \left\| f^{(\alpha)} \right\|_\infty < L \right\}.$$

The generalization of $\mathcal{W}^{s,\infty}$ for a non-integer $s$ results in the space of Hölder-Zygmund type functions. For a detailed exposition of such functions, one may turn to [31].

We consider $\mu$ and $\nu$ to have corresponding densities $p_\mu$ and $p_\nu$, with respect to Lebesgue measures, in their respective spaces. The following assumption provides coherence to their characterization.

**Assumption 1.** *(Regularity of distributions) There exists $m_x, m_y \in \mathbb{N}^+$ such that $p_\mu \in \mathcal{W}_L^{m_x,p}(\Omega_x)$ and $p_\nu \in \mathcal{W}_L^{m_y,q}(\Omega_y)$, where the supports $\Omega_x \subseteq \mathbb{R}^d$ and $\Omega_y \subseteq \mathbb{R}^k$ are both compact, $p, q \in [1, \infty)$.*

Feature extracted image data, in vectorized form, tend to hail from bounded domains in each of its coordinates. Our characterization of input laws having compact support complements this fact. Perhaps it is the very reason that motivates [21] and [20] to assume the same, contextually.

## 4.2 Class of discriminator functions

Functional classes $\mathscr{L}_X, \mathscr{L}_Y$ are characterized based on their ability to tell apart real and generated observations. Some of the notable choices of the same include functions defined over a Reproducing Kernel Hilbert Space (RKHS) [32], Lipschitz [16], and Sobolev functions [33]. In our work, we concentrate on two families, namely $\mathscr{L}_c^1$ and $\mathcal{W}^{m,\infty}$. Our first choice is motivated by the heightened generative quality the Wasserstein distance brings along to deep models. It also offers a pathway to fend off mode collapse and vanishing gradients. On the other hand, the latter class of critics enables us to study the effect of improved smoothness on translation and regeneration.

So far, we have only discussed the Lagrangian formulation of the optimization problem at hand. In fact, (2) is the embodiment of the exact *Lagrange dual function*. The forthcoming analysis, however, relies on the 'constrained version' [Chapter 5 of [34]] given as follows:

$$\inf_{\substack{F \in \mathscr{F}(\mathcal{X}, \mathscr{P}(\mathcal{Y})) \\ G \in \mathscr{F}(\mathcal{Y}, \mathscr{P}(\mathcal{X}))}} \left\{ \mathcal{L}_{cyc}(\mu, \nu, F, G) \right\} \text{ subject to } d_{\mathscr{L}_X}(\mu, G_\# \nu) \leq t_1 \text{ and } d_{\mathscr{L}_Y}(\nu, F_\# \mu) \leq t_2, \quad (3)$$

where $t_1, t_2 \geq 0$. Solutions from (2) turn out to be lower bounds to that derived from (3), a fact that inspires the forthcoming theory. We say 'simultaneous successful translations have taken place' only when the constraints in (3) are met with. The immediate inquiry that follows involves checking the feasibility of an architecture to achieve cycle-consistency. As supporting evidence for both phenomena, we produce deterministic upper bounds on the respective errors along with convergence guarantees of distributions.

## 4.3 Translation guarantees

Let us concentrate on the backward translation ($\mu \xleftarrow{G} \nu$) first. Observe that, a realized sample counterpart of the objective turns out to be $d_{\mathscr{L}_X}(\hat{\mu}_{n_1}, G_\# \hat{\nu}_{n_2})$, where $\hat{\mu}_{n_1} = \frac{1}{n_1} \sum_{i=1}^{n_1} \delta_{X_i}$ is the empirical distribution corresponding to $\mu$, based on $n_1 \in \mathbb{N}^+$ i.i.d. samples $\{X_i\}_{i=1}^{n_1}$. Similarly, $\hat{\nu}_{n_2}$ stands for the same in case of $\nu$, given $n_2 \in \mathbb{N}^+$ samples. As a consequence, any backward translator $G \in \mathscr{F}(\mathcal{Y}, \mathscr{P}(\mathcal{X}))$ should be recognized as $G(n_1, n_2)$. Non-uniqueness of the members residing in the kernel of CycleGAN loss is a well-known fact [9]. Our goal is to prescribe real architectures that induce maps satisfying the first constraint in the sample version of (3). Such recommendations rely on the next definition.

**Definition 4** (ReLU Neural Network)**.** *Given $L \in \mathbb{N}^+$, a $L$-deep Neural Network (NN) is defined as the collection of maps $\phi : \mathbb{R}^{N_0} \longrightarrow \mathbb{R}^{N_{L+1}}$, $\{N_i\}_{i=0}^{L+1} \in \mathbb{N}^+$ given by*

$$\phi(x) := A_L \circ \sigma \circ A_{L-1} \circ ... \circ \sigma \circ A_0(x),$$

*where $A_i(y) = M_i y + b_i$; $M_i \in \mathbb{R}^{N_{i+1} \times N_i}$ and $b_i \in \mathbb{R}^{N_{i+1}}$, $i = 0, ..., L$ is an affinity. The activation $\sigma(y) = y \vee 0$, $y \in \mathbb{R}$. Under this setup, we call $W = \vee_{i=1}^L N_i$ the width of the network. Denote this collection by $\Phi(W, L)_{N_0}^{N_{L+1}}$.*

It is fair to say that ReLU is the most commonly used activation function in modern deep NNs. It is simple to use and also speeds up training, much to practitioners' delight. Moreover, it is superior at dealing with vanishing gradients compared to $sigmoid$ or $tanh$. However, we draw inspiration from the remarkable approximation capability that ReLU-based networks offer [35, 36], especially towards smooth functions [37].

**Theorem 1.** *There exist backward translators $\phi$, based on ReLU neural network $\Phi(W, L)_k^d$ with width $W \geq 7d + 1$ and depth $L \geq 3$, such that whenever $n_1 \leq \frac{W-d-1}{2} \lfloor \frac{W-d-1}{6d} \rfloor \lfloor \frac{L}{2} \rfloor + 2$, we have*

$$\mathbb{E}\left[d_{\mathscr{L}_c^1}(\hat{\mu}_{n_1}, \phi_\# \hat{\nu}_{n_2})\right] \precsim (k^2 n_2)^{-\frac{1}{k}} + \sqrt{k} W^{-\frac{2}{k}} L^{-\frac{2}{k}}.$$

*Proof of Theorem 1.* Let us begin by fragmenting the translation error as the following,

$$d_{\mathscr{L}_c^1}(\hat{\mu}_{n_1}, \phi_\# \hat{\nu}_{n_2}) \leq d_{\mathscr{L}_c^1}(\hat{\mu}_{n_1}, \phi_\# \nu) + d_{\mathscr{L}_c^1}(\phi_\# \nu, \phi_\# \hat{\nu}_{n_2}). \tag{4}$$

Before moving forward, denote the set of discrete probability measures based on at most $n \in \mathbb{N}^+$ points in $\mathbb{R}^d$ by, $\mathcal{P}_n(d) := \left\{ \sum_{i=1}^n a_i \delta_{x_i} : a_i \geq 0, \sum_{i=1}^n a_i = 1, \{x_i\}_{i=1}^n \in \mathbb{R}^d \right\}$. The following lemma allows us to show that the first term on the right-hand side of (4) can be made arbitrarily small.

**Lemma 1** ([38]). *Let $p$ be an absolutely continuous univariate distribution and $\pi \in \mathcal{P}_n(d)$. There exists $\phi \in \Phi(W, L)_1^d$ with $W \geq 7d + 1$ and $L \geq 2$, such that whenever $n \leq \frac{W-d-1}{2} \lfloor \frac{W-d-1}{6d} \rfloor \lfloor \frac{L}{2} \rfloor + 2$*

$$d_{\mathscr{L}_c^1}(\pi, \phi_\# p) \leq \epsilon, \quad given \ \epsilon > 0.$$

Observe that, $\hat{\mu}_{n_1} \in \mathcal{P}_{n_1}(d)$. Now, choose $\phi \in \Phi(W, L)_k^d$ such that the first layer deploys an additional linear map $A$ that projects $\nu$ to a one-dimensional absolutely continuous distribution first. This can always be done due to the absolute continuity of $\nu$ itself. For the resultant map $\phi$, we notice $L \geq 3$, and the specifications of $W, n_1$ remain as directed by lemma (1). As a result, $d_{\mathscr{L}_c^1}(\hat{\mu}_{n_1}, \phi_\# \nu) \leq \epsilon$ for arbitrary $\epsilon > 0$.

The second term in (4) is the portion of the density estimation error from the base domain that translates on to the target space. To get control over such a discrepancy we exploit the regularity of the transformation $\phi$. Observe that, the activation function $\sigma \equiv \text{ReLU}$ is 1-Lipschitz. The transformation carrying signal $y$ from $i^{th}$ layer to the next is of the form $A_i(y) = M_i y + b_i$; $M_i \in \mathbb{R}^{N_{i+1} \times N_i}$ and $b_i \in \mathbb{R}^{N_{i+1}}$, $i = 0, ..., L$. The matrix $M_i$ can be constructed such that $\|M_i\|_p = \sup_{\|y\|_p = 1} \|M_i y\|_p \leq k_i$, for some constant $k_i > 0$. For cases $p = 2$ or $\infty$, [39] present exact techniques to ensure $\|M_i\|_p = 1$. Under such a framework, $A_i$'s become $k_i$-Lipschitz transforms. Since Lipschitz functions are closed under composition, we can expect $\phi$ to behave similarly, with a constant $k^*$ dependent on $\{k_0, k_1, ..., k_L\}$.

However, deep ReLU networks are much more expressive and are capable of approximating a vast array of smooth functions. Let us denote the class of $L_G$-Lipschitz functions mapping $(\Omega_y, c') \rightarrow (\Omega_x, c)$ as $G_{Lip}$, where $L_G > 0$. The following two results encapsulate our idea precisely.

**Lemma 2.** *For $\alpha, \beta \in \mathscr{P}(\mathcal{Y})$, $d_{\mathscr{L}_c^1}(\phi_\# \alpha, \phi_\# \beta) \leq 2 \inf_{g \in G_{Lip}} \|\phi - g\|_\infty + L_G \, d_{\mathscr{L}_{c'}^1}(\alpha, \beta)$.*

**Lemma 3** ([40]). *Let $g \in G_{Lip}$. Also, $\phi$ is the ReLU NN-induced function as given in Lemma (1), having width $\mathcal{O}(W)$ and depth $\mathcal{O}(L)$. Then*

$$\|\phi - g\|_\infty \leq \mathcal{O}(C_1 W^{-\frac{2}{k}} L^{-\frac{2}{k}}),$$

*where $C_1 > 0$ is a constant, dependent on $L_G$, $\sqrt{k}$, and $diam(\Omega_y)$.*

Using both lemmas, we obtain $d_{\mathscr{L}_c^1}(\hat{\mu}_{n_1}, \phi_\# \hat{\nu}_{n_2}) \leq \epsilon + L_G \, d_{\mathscr{L}_{c'}^1}(\nu, \hat{\nu}_{n_2}) + \mathcal{O}(C_1 W^{-\frac{2}{k}} L^{-\frac{2}{k}})$. The sole task remaining is to upper bound the statistical estimation error in the base space. To that end, by applying Corollary 2.1 of [41] for $k \geq 2$, we get $\mathbb{E}_\nu[d_{\mathscr{L}_{c'}^1}(\nu, \hat{\nu}_{n_2})] \leq \mathcal{O}((k^2 n_2)^{-\frac{1}{k}})$. $\qed$

This result enables us to formally present what we mean by 'translation guarantee'. The next corollary can be seen as an embodiment of the same idea.

**Corollary 1** (Translation consistency). *As $\min(n_1, n_2) \rightarrow \infty$, we have $d_{\mathscr{L}_c^1}(\hat{\mu}_{n_1}, \phi_\# \hat{\nu}_{n_2}) \xrightarrow{a.s.} 0$.*

In other words, given sufficient information from both the distributions, the backward translation method governed by a map $\phi$ satisfies the constraint in the sample version of (3). The corollary is an asymptotic statement that ensures the error eventually shrinks below any given $t_1 > 0$. A crucial observation in this context is that for $m \geq 1$, $\mathcal{W}_1^{m,\infty}$ is a sub-family of bounded Lipschitz functions. In our case, since the supports of the distributions are taken to be bounded, one may equivalently say $\mathcal{W}_1^{m,\infty} \subset \mathcal{L}_c^1$, $c \equiv L^1$. As such, $\mathcal{W}_1^{m,\infty}$ playing the role of the critic should produce results similar to Theorem (1). From this point onward, all proofs are placed in the supplementary material.

**Theorem 2.** *For a backward translator $\phi$ of width $W$ and depth $L$, as specified in Theorem (1)*

$$\mathbb{E}\big[d_{\mathcal{W}_1^{m,\infty}}(\hat{\mu}_{n_1}, \phi_\# \hat{\nu}_{n_2})\big] \precsim n_2^{-\frac{m}{k}} + \frac{\log n_2}{\sqrt{n_2}} + \sqrt{k} W^{-\frac{2}{k}} L^{-\frac{2}{k}},$$

*where $n_1 \leq \frac{W-d-1}{2}\lfloor \frac{W-d-1}{6d} \rfloor \lfloor \frac{L}{2} \rfloor + 2$ and $n_2 \in \mathbb{N}^+$.*

One might wonder what makes Lipschitz transformations so relevant to this context. The first rather evident observation is that it restricts any further amplification of the distance between laws post-translation. The next reason, a particular consequence of the former, brings us to the concept of *Information preserving transformations* (IPT) [11]. A contextual definition of the same is as follows

**Definition 5** (IPT [11]). *A map $I \in \mathscr{F}(\mathcal{Y}, \mathscr{P}(\mathcal{X}))$ is said to be an Information preserving transformation of degree $r \geq 1$ under distance metric $d$, if there exist constants $k_1, k_2 \geq 0$, such that*

$$\mathbb{P}\Big(d\big(I_\# \hat{\nu}_n, \widehat{(I_\# \nu)}_n\big) \leq \epsilon\Big) \geq 1 - k_1 \exp\left(-k_2 n^r \epsilon^2\right).$$

Here, $\widehat{(I_\# \nu)}_n$ is an empirical counterpart of the translated law $I_\# \nu$ based on $n \in \mathbb{N}^+$ samples. As such, IPTs are maps that ensure the error committed while replacing $I_\# \hat{\nu}_n$ with $\widehat{(I_\# \nu)}_n$ (information dissipated) remains arbitrarily small, with a high probability. The next result suggests that Lipschitz-regular transforms behave as IPT when the target class of distributions is not too 'complex'.

**Definition 6** (Yatracos family [42]). *Given a class of functions $\mathcal{F} : \Omega \to \mathbb{R}$, the Yatracos family associated to it is defined as,*

$$\mathcal{Y}(\mathcal{F}) = \{\omega \in \Omega : f(\omega) \geq g(\omega); \ f, g \in \mathcal{F}\}.$$

**Proposition 1** (Information preservation of Lipschitz translators). *Let the Vapnik–Chervonenkis (VC) dimension of $\mathcal{Y}(\mathscr{P}(\mathcal{X}))$ be finite. Then for any $g \in G_{Lip}$, there exists a constant $C_2 > 0$ such that*

$$\mathbb{P}\Big(d_{\mathcal{L}_c^1}(g_\# \hat{\nu}_n, \widehat{(g_\# \nu)}_n) \leq t + \mathcal{O}(n^{-\frac{1}{k \vee 2}})\Big) \geq 1 - 2\exp\left(-C_2 n t^2\right).$$

**Remark 3.** Observe that, maps induced by ReLU feed-forward networks can similarly pose as IPT, incurring an additional approximation error of order $\mathcal{O}(W^{-\frac{2}{k}} L^{-\frac{2}{k}})$ [lemma (2)]. This near-perfect behaviour makes the choice of ReLU-NNs, as suitable translators, rather inevitable. However, neural networks based on $tanh$ [43], $sigmoid$ [44], and GroupSort [45] activations have also been shown to approximate Lipschitz functions with high precision. We feel, a comparative analysis of the activations based on their effectiveness in the face of information dissipation may lead to improved prescriptions.

**Remark 4** (Forward translation). We stress the fact that so far in our discussion, the data dimensions $d, k$ have no restrictions put on them. As such, the same arguments hold true for the forward generation process ($\mu \xrightarrow{F} \nu$) as well. A forward translator map $\psi \in \Phi(W, L)_d^k$ can be similarly constructed that achieves translation consistency. In other words, one may easily check that the second constraint in (3) is also satisfied for arbitrary values of $t_2$.

Cycle-consistent networks find themselves under the obligation to reconstruct the input signal following their translation. A major obstacle in the process, however, that often mars the quality of regenerated observations is ill-posedness. It stems from a translation belonging to the feasible set of solutions that results in noisy output, devoid of sufficient information to aid the reconstruction. Theoretically, the remedy to ill-posedness lies in the formation of a 'perfect' transport map between the measures. Having IPT (Lipschitz) as a reference, NN-based transports tend to overcome this issue asymptotically [Corollary (1)]. However, measurable maps, in general, lack such approximation capability. Our following discussion sheds light on the same.

For this section, let us assume the dimensions of the two data domains to be equal, i.e., $d = k$. This occasion, in particular, has interesting consequences. Given that $p_\mu$ and $p_\nu$ have finite variance, Brenier's theorem [46] ensures the existence of a unique solution $\gamma = (Id \times T)_{\#}\nu$ to the Kantotovich Optimal Transport (OT) problem: $W^1_{c \equiv L_2}$. In other words, we get hold of a map $T \in \mathscr{F}(\mathcal{Y}, \mathscr{P}(\mathcal{X}))$ such that $T_{\#}\nu = \mu$. It is clear that any function aspiring to subdue ill-posedness should lie in an $\epsilon$-envelope of $T$, $\epsilon > 0$ being as small as possible. The larger the deviation, the more is the extent of degradation in reconstructed image quality. Drawing inspiration from this fact, [47, 13] deploy OT-based regularizers to guide the solution map toward $T$. However, the regularity properties of $T$ can only be determined under very specific assumptions on the data domains [48, 49]. Moreover, we only have access to an empirical counterpart of the target law in the sample version of the problem. As a result, approximations of the transport map are likely to be noisy. Our next result aims at pointing out the tolerable error margin due to ill-posedness in a sample backward translation.

**Lemma 4.** *For a discriminator class $\mathscr{L}_X$, and a backward translator $G$*

$$d_{\mathscr{L}_X}(\hat{\mu}_{n_1}, G_{\#}\hat{\nu}_{n_2}) \leq \mathcal{E}_1 + \mathcal{E}_2 + \mathcal{E}_3,$$

*where $\frac{\mathcal{E}_1}{B_x} := \left\|\hat{\mu}_{n_1} - T_{\#}\nu\right\|_{TV}$ (Statistical approximation error in target space),*
$\mathcal{E}_2 := B_x \left\|\Gamma_{n_1} - \widehat{(G_{\#}\nu)}_{n_2}\right\|_{TV} = \Lambda_{(n_1,n_2)}$; *given* $\Gamma_{n_1} = \text{argmin}_{\tau \in \mathscr{P}(\mathcal{X})} \|\tau - \hat{\mu}_{n_1}\|_{TV}$, $B_x = diam(\Omega_x)$ *with respect to the metric $c$, and $\mathcal{E}_3 := d_{\mathscr{L}_X}\left(\widehat{(G_{\#}\nu)}_{n_2}, G_{\#}\hat{\nu}_{n_2}\right)$ (Information dissipated).*

The quantity $\Lambda_{(n_1,n_2)}$ represents the cost incurred by $\widehat{(G_{\#}\nu)}_{n_2}$ for partaking in the Scheffe tournament [42] to approach $\hat{\mu}_{n_1}$. Observe that, it remains an admissible amount of deviation if $\lim_{\min(n_1,n_2)\to\infty} \Lambda_{(n_1,n_2)} \leq t_1$ (3). The maps, ensuring $\lim_{\min(n_1,n_2)\to\infty} \Lambda_{(n_1,n_2)} = 0$, belong to the set of 'pure' solutions [9]. Theoretically, the negative effect of such maps on the regeneration quality would be benign. We elaborate on the same in Proposition (2). Also, observe that Lemma (4) re-emphasizes the necessity of a backward translator to be an IPT.

**Remark 5** (Mode collapse). In real situations, supports of data distributions at both ends are often non-convex. This is an important feature that makes OT maps ($T$) discontinuous [50]. On the other hand, neural networks lack proficiency in approximating such discontinuous functions. For multi-modal input laws, an estimated transformation approximating only the continuous branches of the target OT map results in mode collapse during translation [50]. As such, the error associated with mode collapse remains convoluted in $\mathcal{E}_2$. Fragmenting the realized estimation loss into finer components to address mode collapse may be taken up as potential future work.

Let us now shift our focus towards the residual task a cycle-consistent I2I translator needs to execute.

## 4.4 Cycle consistency analysis

The cyclic loss, as given in (2), measures the expected discrepancy between the input data and its reconstructed counterpart. However, the density estimation approach we follow allows us to reframe the objective as a divergence between distributions, given as

$$\mathcal{L}_{cyc}(\mu, \nu, F, G) = \left\|\mu - (G \circ F)_{\#}\mu\right\|_1 + \left\|\nu - (F \circ G)_{\#}\nu\right\|_1.$$

For two distributions $P, Q$; $\|P - Q\|_1 = 2\|P - Q\|_{TV} = \int |\rho_P - \rho_Q| d\lambda$, given that $\frac{dP}{d\lambda} = \rho_P$ and $\frac{dQ}{d\lambda} = \rho_Q$. This formulation provides a stronger notion of the loss. The first result of this section discovers the relationship between translation and reconstruction.

**Lemma 5.** *For $G \in \mathscr{F}(\mathcal{Y}, \mathscr{P}(\mathcal{X}))$ and $F \in \mathscr{F}(\mathcal{X}, \mathscr{P}(\mathcal{Y}))$,*

$$\mathcal{L}_{cyc}(\mu, \nu, F, G) \leq 2\left\{\left\|\mu - G_{\#}\nu\right\|_1 + \left\|\nu - F_{\#}\mu\right\|_1\right\}.$$

The extent to which a cycle consistent translator can be inaccurate is determined by its performance in the simultaneous generations. This is a rather desired outcome. However, the indication of much intrigue that this result gives is that in case the translations are 'successful' in both directions, cycle-consistency can be achieved. One key feature of the input distributions that become crucial henceforth is their smoothness. As a consequence of Corollary (1), we infer $\phi_{\#}\hat{\nu}_{n_2} \to \mu$ weakly. A similarly constructed forward translator $\psi$ may also ensure that $\psi_{\#}\hat{\mu}_{n_1} \to \nu$ weakly. Based on such guarantees, we make the following assumptions about the regularity of the transported laws.

**Assumption 2.** *The translated distributions $\phi_{\#}\nu$ and $\psi_{\#}\mu$ possess corresponding densities given by $p_{\phi_{\#}\nu} \in \mathcal{W}_L^{m_x,p'}(\Omega_x)$ and $p_{\psi_{\#}\mu} \in \mathcal{W}_L^{m_y,q'}(\Omega_y)$; $p', q' \in [1,\infty)$.*

Before presenting upcoming theoretical results, let us introduce a tool that plays a key role henceforth.

**Definition 7** (Regularly Invariant Kernels [31]). *A measurable function $K(x,y) : \mathbb{R}^d \times \mathbb{R}^d \longrightarrow \mathbb{R}$ is said to be 'Regular', if for $N \in \mathbb{N}$*

*1. $\int_{\mathbb{R}^d} \sup_{v \in \mathbb{R}^d} |K(v, v-u)| |u|^N du < \infty$,*

*2. for every $v \in \mathbb{R}^d, |\alpha| = 1, ..., N-1$ we have $\int_{\mathbb{R}^d} K(v, v+u)du = 1$ ; $\int_{\mathbb{R}^d} K(v, v+u)u^\alpha du = 0$. If such a kernel also satisfies the 'Invariance' property:*

$$\left\{ \int \left| K(w,v) - K(w,u) \right|^r dw \right\}^{\frac{1}{r}} = \mathcal{O}(|v-u|),$$

*for $r \geq 1$, we call it 'Regularly Invariant'.*

We mention that the total variation metric can also be expressed as a transportation distance, the underlying cost function being $c(x,y) = 1_{x \neq y}$. However, as Chae *et al.* [51] point out, the topologies that the TV and Wasserstein distances generate are hardly comparable. For Sobolev densities, TV often fails to appreciate the nuances that 'smoothness' brings along. A method of alleviation of such difficulties lies in regular kernels. Minute deviations between smooth functions can be apprehended in greater detail when convoluted with such kernels. Inevitably, regularly invariant kernels become the cornerstone of our next result. The proof, placed in the supplement, highlights its contribution.

**Theorem 3.** *Given the metric $c \equiv L^1$, there exists a constant $M > 0$ dependent on $m_x$, such that*

$$\left\| p_\mu - p_{\phi_{\#}\nu} \right\|_1 \leq M \left[ \left\| D^{m_x} p_\mu \right\|_p + \left\| D^{m_x} p_{\phi_{\#}\nu} \right\|_{p'} \right]^{\frac{1}{m_x+1}} \left[ d_{\mathcal{L}_c^1}(\mu, \phi_{\#}\nu) \right]^{\frac{m_x}{m_x+1}}.$$

**Remark 6.** This result is a multivariate generalization of Theorem 2.1 in [51].

Note that, a similar conclusion can also be drawn for the loss indicating the difference between the target and generated density in case of forward translation. That is to say,

$$\left\| p_\nu - p_{\psi_{\#}\mu} \right\|_1 \leq M' \left[ \left\| D^{m_y} p_\nu \right\|_q + \left\| D^{m_y} p_{\psi_{\#}\mu} \right\|_{q'} \right]^{\frac{1}{m_y+1}} \left[ d_{\mathcal{L}_c^1}(\nu, \psi_{\#}\mu) \right]^{\frac{m_y}{m_y+1}}, \tag{5}$$

where $M'$ is a constant depending on $m_y$. Likewise, any pair of translators $(G, F)$ that preserve the smoothness of input densities onto generated ones satisfy Theorem (3) and (5). The collective evidence from these two results suggest that a sufficient condition for achieving cycle-consistency is the arbitrary closeness between real and translated Sobolev-smooth densities, in both domains, under the 1-Wasserstein metric. Moreover, we already know that $\frac{2}{B_x} d_{\mathcal{L}_c^1}(\mu, \phi_{\#}\nu) \leq \left\| \mu - \phi_{\#}\nu \right\|_1$ [52]. As such, establishing translation consistency under the critic $\mathcal{L}_c^1$ is equivalent to attaining cycle-consistency.

Now, let us focus on the sample version of the cyclic loss, given as $\mathcal{L}_{cyc}(\hat{\mu}_{n_1}, \hat{\nu}_{n_2}, F, G)$. The inability of translation maps to approximate optimal transports up to arbitrarily high accuracy affects cycle-consistency as well. Noisy outputs from a backward generation process should not ideally recover, even under a 'perfect' forward translator. Meanwhile, a 'perfectly' translated forward image will be distorted due to such imperfect backward generators. If the effects due to the departure of translator maps from their 'ideal' benchmarks get multiplied, we may observe severe corruption in reconstruction quality. Much to our relief, the next result assures that the effects of ill-posedness amplify only as a sum.

**Proposition 2.** *Denote $B_y \left\| \Gamma'_{n_2} - \widehat{(F_{\#}\mu)}_{n_1} \right\|_{TV} = \Lambda'_{(n_1,n_2)}$, given that $\Gamma'_{n_2} = \mathrm{argmin}_{\tau \in \mathscr{P}(\mathcal{Y})} \|\tau - \hat{\nu}_{n_2}\|_{TV}$, $B_y = diam(\Omega_y)$ with respect to the metric $c'$. Then*

$$\mathcal{L}_{cyc}(\hat{\mu}_{n_1}, \hat{\nu}_{n_2}, F, G) - 4 \left\{ \frac{\Lambda_{(n_1,n_2)}}{B_x} + \frac{\Lambda'_{(n_1,n_2)}}{B_y} \right\} \leq \mathcal{E}_1^* + \mathcal{E}_2^*,$$

*where $\mathcal{E}_1^* := 4 \left\{ \|\hat{\mu}_{n_1} - \mu\|_{TV} + \|\hat{\nu}_{n_2} - \nu\|_{TV} \right\}$ (Cumulative statistical approximation error), $\mathcal{E}_2^* := 4 \left\{ \left\| \widehat{(F_{\#}\mu)}_{n_1} - F_{\#}\hat{\mu}_{n_1} \right\|_{TV} + \left\| \widehat{(G_{\#}\nu)}_{n_2} - G_{\#}\hat{\nu}_{n_2} \right\|_{TV} \right\}$ (Total information dissipated).*

It is expected of the pair of maps that commit zero translation error (e.g. $(\phi, \psi)$, asymptotically) to belong to the 'kernel' of a cycle-consistent network. In other words, the realized cyclic loss should also lie near zero. To showcase the idea of reconstruction consistency let us concentrate on the term:

$$\hat{\mathcal{L}}_{cyc}(\hat{\mu}_{n_1}, \hat{\nu}_{n_2}, \psi, \phi) = \left\|\mu - (\phi \circ \psi)_{\#}\hat{\mu}_{n_1}\right\|_1 + \left\|\nu - (\psi \circ \phi)_{\#}\hat{\nu}_{n_2}\right\|_1.$$

Since the smoothness of underlying distributions is paramount in our analysis, usage of regularly invariant kernel density estimates $(\tilde{\mu}_{n_1}, \tilde{\nu}_{n_2})$ instead may lead to improved approximation. Based on the same set of observations we build $\hat{p}_{\mu,n_1}(x) = \frac{d\tilde{\mu}_{n_1}}{dx} = \frac{1}{n_1 h^d}\sum_{i=1}^{n_1} K\left(\frac{x}{h}, \frac{x_i}{h}\right)$, $x \in \Omega_x$ where $h \equiv h(n_1)$. Similarly define $\hat{p}_{\nu,n_2}$.

**Theorem 4.** *For the pair of forward-backward maps $(\psi, \phi)$, as constructed in Theorem (1)*

$$\mathbb{E}\left[\hat{\mathcal{L}}_{cyc}(\tilde{\mu}_{n_1}, \tilde{\nu}_{n_2}, \psi, \phi)\right] \precsim \max\left\{n_1^{-\frac{m_x}{(d \vee 2)m_x + d}}, n_2^{-\frac{m_y}{(k \vee 2)m_y + k}}\right\}.$$

The eventual nullification of the average reconstruction loss is a desirable outcome. However, the concentration of random empirical losses around such an aggregate bears more significance. On that note, we present the concluding result that embodies our idea of reconstruction consistency.

**Corollary 2** (Regeneration consistency). *As $\min(n_1, n_2) \to \infty$, we observe $(\phi \circ \psi)_{\#}\tilde{\mu}_{n_1} \to \mu$ and $(\psi \circ \phi)_{\#}\tilde{\nu}_{n_2} \to \nu$, both in total variation.*

**Remark 7.** While the usage of 'smoother' estimates produce faster convergence rates, usual empirical distributions $(\hat{\mu}_{n_1}, \hat{\nu}_{n_2})$ also lead to an outcome similar to Corollary (2), given that the VC dimensions of both $\mathcal{Y}(\mathscr{P}(\mathcal{X}))$ and $\mathcal{Y}(\mathscr{P}(\mathcal{Y}))$ are finite.

## 5 Conclusion and Future Work

This study establishes statistical translation and regeneration guarantees of cycle-consistent networks. In the process, we recommend precise recipes to build translator maps to achieve such consistency. To the best of our knowledge, it is the first endeavour of its kind in this context. We prove that deep ReLU-based translators, being fine approximators of Lipschitz functions, asymptotically behave like IPTs. We theoretically show that for Sobolev-smooth input data, deployment of the 1-Wasserstein distance and $L^1$ in the cyclic loss are equivalent. This substantiates the conclusion Zhu *et al.* [2] had reached for CycleGAN. A key highlight of our analysis is the absence of any restrictions on the data dimensions. We also discuss the ramifications of ill-posedness during translation and the impact it leaves on the regeneration. The decomposition of the translation and cyclic errors in the process, based on independent sources of variation, is also new in this setting.

Our analytical approach may pave the way for further scrutiny of the many unexplored areas of cycle-consistent networks.

**Robustness:** Deep generative models, in general, are often found to be vulnerable in the presence of outliers in input data. Required is a study on the tolerable proportion of extreme values from both domains that cycle-consistent I2I translators can handle. Future work may also look into the robustness of such networks, given that the translation maps have a disjoint noisy component with thick tails. Moreover, such networks are found to be prone to self-attacking. In case the target mapping is many-to-one (e.g. photos to semantic labels), the realized translators tend to hide information as a noisy component in the translated law, imperceptible to discriminators. Though effective defense mechanisms against self-attack (adversarial training with noise, and using guess discriminators) have been proposed [53], deterministic bounds on the permissible departure of maps from their theoretical references remain absent.

**Training:** Another aspect that lies beyond the scope of this article is the training process. In our non-parametric approach, we do not recognize the set of candidate distributions to be exactly characterized by an underlying parameter space (say, $\Theta$). On the other hand, training can be viewed as the process of finding out a suitable parameter value (say, $\hat{\theta} \in \Theta$) such that the corresponding density estimate optimizes the loss. This is crucial since, during training, the parameters of the translator networks are responsible for shaping up this parameter space, and hence the optimum. In this regime, the optimization trajectory can accordingly be viewed as the stochastic propagation of $\hat{\theta}$ (as a function of the input sample size, and iterations or time) towards a stable value. Hence, we feel a parametric analysis with a spirit similar to ours can serve the questions rooted in training and related optimization justice.

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
