# Supplement to : 'On Translation and Reconstruction Guarantees of the Cycle-Consistent Generative Adversarial Networks'

**Anish Chakrabarty**
Statistics and Mathematics Unit
Indian Statistical Institute, Kolkata
West Bengal, India

**Swagatam Das**
Electronics and Communication Sciences Unit
Indian Statistical Institute, Kolkata
West Bengal, India

## Appendix

***Proof of Lemma (2).*** Let us begin by specifying the class of discriminators $\mathscr{L}_X \equiv \mathscr{L}_c^1$. Now, given $\alpha, \beta \in \mathscr{P}(\mathcal{Y})$

$$d_{\mathscr{L}_X}(\phi_{\#}\alpha, \phi_{\#}\beta) = \sup_{l \in \mathscr{L}_X} \left[ \mathbb{E}_{\phi_{\#}\alpha} l - \mathbb{E}_{\phi_{\#}\beta} l \right] = \sup_{l \in \mathscr{L}_X} \left[ \mathbb{E}_{\alpha}(l \circ \phi) - \mathbb{E}_{\beta}(l \circ \phi) \right].$$

Due to the definition of supremum, for any $\epsilon > 0 \; \exists \; l_{\epsilon} \in \mathscr{L}_X$ for which

$$d_{\mathscr{L}_X}(\phi_{\#}\alpha, \phi_{\#}\beta) \le \mathbb{E}_{\alpha}(l_{\epsilon} \circ \phi) - \mathbb{E}_{\beta}(l_{\epsilon} \circ \phi) + \epsilon$$

$$= \inf_{g \in l_{\epsilon} \circ G_{Lip}} \left\{ \mathbb{E}_{\alpha} |(l_{\epsilon} \circ \phi) - g| - \mathbb{E}_{\beta} |(l_{\epsilon} \circ \phi) - g| + \mathbb{E}_{\alpha}(g) - \mathbb{E}_{\beta}(g) \right\} + \epsilon$$

$$\le 2 \inf_{g' \in G_{Lip}} \left\| \phi - g' \right\|_{\infty} + \left\{ \sup_{l \in \mathscr{L}_X} \left[ \mathbb{E}_{\alpha}(l \circ g^*) - \mathbb{E}_{\beta}(l \circ g^*) \right] \right\} + \epsilon, \; \forall \; g^* \in G_{Lip}.$$

Here, $l_{\epsilon} \circ G_{Lip} := \{l_{\epsilon} \circ f : f \in G_{Lip}\}$. Now,

$$\sup_{l \in \mathscr{L}_X} \left[ \mathbb{E}_{\alpha}(l \circ g^*) - \mathbb{E}_{\beta}(l \circ g^*) \right] = \inf_{\gamma \in \Gamma(\alpha, \beta)} \int c\big(g^*(x), g^*(y)\big) d\gamma(x, y)$$

$$\le L_G \inf_{\gamma \in \Gamma(\alpha, \beta)} \int c'(x, y) d\gamma(x, y), \tag{1}$$

where (1) is due to the fact that $g^* \in G_{Lip}$. As such,

$$d_{\mathscr{L}_c^1}(\phi_{\#}\alpha, \phi_{\#}\beta) \le 2 \inf_{g' \in G_{Lip}} \left\| \phi - g' \right\|_{\infty} + L_G \, d_{\mathscr{L}_{c'}^1}(\alpha, \beta).$$

$\square$

***Proof of Corollary (1).*** We have already noticed $\mathbb{E}_{\nu}[d_{\mathscr{L}_{c'}^1}(\nu, \hat{\nu}_{n_2})] \le \mathcal{O}((k^2 n_2)^{-\frac{1}{k}}), k \ge 2$. Since the distance $d_{\mathscr{L}_{c'}^1}(., .)$ satisfies the bounded difference inequality, the application of McDiarmid's inequality leads to

$$\mathbb{P}\Big(d_{\mathscr{L}_{c'}^1}(\nu, \hat{\nu}_{n_2}) \le \mathcal{O}((k^2 n_2)^{-\frac{1}{k}}) + t\Big) \ge 1 - \exp\left\{ -\frac{2n_2 t^2}{B_y^2} \right\}, \tag{2}$$

where $B_y = diam(\Omega_y)$ with respect to the metric $c'$. We point out that (2) is a generalized version of Proposition 20 in [1]. Now, Theorem (1) tells us,

$$d_{\mathscr{L}_c^1}(\hat{\mu}_{n_1}, \phi_{\#}\hat{\nu}_{n_2}) \le \epsilon + L_G \, d_{\mathscr{L}_{c'}^1}(\nu, \hat{\nu}_{n_2}) + \mathcal{O}(C_1 W^{-\frac{2}{k}} L^{-\frac{2}{k}}),$$

36th Conference on Neural Information Processing Systems (NeurIPS 2022).

given $\epsilon > 0$ and $n_1 \le \frac{W-d-1}{2}\lfloor\frac{W-d-1}{6d}\rfloor\lfloor\frac{L}{2}\rfloor + 2$. Combining these two results, we get

$$\mathbb{P}\Big(d_{\mathscr{L}_c^1}(\hat{\mu}_{n_1}, \phi_\#\hat{\nu}_{n_2}) \le \mathcal{O}((k^2 n_2)^{-\frac{1}{k}}) + \frac{(1+L_G)B_y}{\sqrt{2}}n_2^{-\frac{1}{2}}\sqrt{\ln\Big(\frac{1}{\delta}\Big)} + \mathcal{O}(C_1 W^{-\frac{2}{k}}L^{-\frac{2}{k}})\Big) \ge 1-\delta,$$

by taking $\delta = \exp\Big\{-\frac{2n_2 t^2}{B_y^2}\Big\}$. The statement also holds if we replace the two sample sizes $n_1, n_2$ with $\min(n_1, n_2)$. In such a case, the Borel-Cantelli lemma implies that $d_{\mathscr{L}_c^1}(\hat{\mu}_{n_1}, \phi_\#\hat{\nu}_{n_2}) \longrightarrow 0$ almost surely (under $\mathbb{P}$), provided $d, k$ remain fixed. □

**Remark.** We draw the attention of the reader to a particular consequence of this result. Observe that the width ($W$) and depth ($L$) of the translator network are intrinsically related to the sample size ($n_1$) from the target law. In case $\min(n_1, n_2) \longrightarrow \infty$, $W$ also follows suit, given that $L$ remains constant. As such, our ideal backward translator, achieving generation consistency, is a finite sample approximation of an infinitely wide ReLU network. Maps induced by such an infinitely wide network converge in distribution to a Gaussian process [2]. This determines the large sample property of $\phi$. Finding out the exact statistical properties of such a process in a parametric setup might be taken up as future work.

**Remark.** For any $n_1 \in \mathbb{N}^+$, $d_{\mathscr{L}_c^1}(\mu, \phi_\#\hat{\nu}_{n_2}) \le d_{\mathscr{L}_c^1}(\mu, \hat{\mu}_{n_1}) + d_{\mathscr{L}_c^1}(\hat{\mu}_{n_1}, \phi_\#\hat{\nu}_{n_2})$. We have already seen that the second term on the right-hand side of the inequality vanishes eventually [Corollary 1]. Moreover, similar to (2)

$$\mathbb{P}\Big(d_{\mathscr{L}_c^1}(\mu, \hat{\mu}_{n_1}) \le \mathcal{O}((d^2 n_1)^{-\frac{1}{d}}) + t\Big) \ge 1 - \exp\Big\{-\frac{2n_1 t^2}{B_x^2}\Big\}.$$

As a result, $d_{\mathscr{L}_c^1}(\mu, \hat{\mu}_{n_1}) \xrightarrow{a.s.} 0$ (using Borel-Cantelli lemma). Hence, it can be concluded that $\phi_\#\hat{\nu}_{n_2}$ converges weakly to $\mu$ in $\mathscr{P}(\mathcal{X})$ [Theorem 6.9 in [3]].

***Proof of Theorem (2).*** Let us carry out the decomposition of the realized backward translation error, similar to that in Theorem (1).

$$d_{\mathcal{W}_1^{m,\infty}}(\hat{\mu}_{n_1}, \phi_\#\hat{\nu}_{n_2}) \le d_{\mathcal{W}_1^{m,\infty}}(\hat{\mu}_{n_1}, \phi_\#\nu) + d_{\mathcal{W}_1^{m,\infty}}(\phi_\#\nu, \phi_\#\hat{\nu}_{n_2}).$$

Observe that $\mathcal{W}_1^{m,\infty} \subset \mathcal{W}_1^{1,\infty}$, for any positive integer $m$. Also, the class $\mathcal{W}_1^{1,\infty}$ is a dense subset of 1-Lipschitz functions on $\mathcal{X}$. As such, $d_{\mathcal{W}_1^{m,\infty}}(\hat{\mu}_{n_1}, \phi_\#\nu) \le d_{\mathscr{L}_c^1}(\hat{\mu}_{n_1}, \phi_\#\nu) \le \epsilon$, where $\epsilon > 0$ (as in the proof of Theorem (1)).

The remaining approximation error can similarly be upper bound using the same technique. However, it would be far from tight. Let us define a class of functions that help in the pursuit of sharper bounds.

**Definition** (Hölder Space). *For $s \in \mathbb{R}_{>0}$, with $\lfloor s \rfloor$ indicating the largest integer strictly smaller than $s$, the Hölder space of order $s$ is defined as*

$$\mathcal{C}_L^s(\mathbb{R}^d) = \Big\{f \in C_u(\mathbb{R}^d) : \|f\|_{\mathcal{C}^s} \equiv \|f\|_{\mathcal{W}^{\lfloor s \rfloor}} + \sum_{|\alpha|=\lfloor s \rfloor} \sup_{\substack{x \ne y \\ x,y \in \mathbb{R}^d}} \frac{|D^\alpha f(x) - D^\alpha f(y)|}{|x-y|^{s-\lfloor s \rfloor}} < L\Big\}.$$

Now, similar to the proof of Lemma (2), for any $\epsilon' > 0$ $\exists$ $l_{\epsilon'} \in \mathcal{W}_1^{m,\infty}$ such that

$$d_{\mathcal{W}_1^{m,\infty}}(\phi_\#\alpha, \phi_\#\beta) \le \mathbb{E}_\alpha(l_{\epsilon'} \circ \phi) - \mathbb{E}_\beta(l_{\epsilon'} \circ \phi) + \epsilon', \quad \text{where } \alpha, \beta \in \mathscr{P}(\mathcal{Y})$$

$$= \inf_{g \in l_{\epsilon'} \circ G_{Lip}} \Big\{\mathbb{E}_\alpha\big|(l_{\epsilon'} \circ \phi) - g\big| - \mathbb{E}_\beta\big|(l_{\epsilon'} \circ \phi) - g\big| + \mathbb{E}_\alpha(g) - \mathbb{E}_\beta(g)\Big\} + \epsilon'$$

$$\le 2\inf_{g' \in G_{Lip}}\Big\|\phi - g'\Big\|_\infty + \Big\{\sup_{l \in \mathcal{W}_1^{m,\infty}}\big[\mathbb{E}_\alpha(l \circ g^*) - \mathbb{E}_\beta(l \circ g^*)\big]\Big\} + \epsilon', \; \forall \, g^* \in G_{Lip}.$$

$$(3)$$

The first term in (3) is obtained due to the Lipschitz property of $l_{\epsilon'}$. Here,

$$\sup_{l \in \mathcal{W}_1^{m,\infty}}\big[\mathbb{E}_\alpha(l \circ g^*) - \mathbb{E}_\beta(l \circ g^*)\big] = d_{\mathcal{W}_1^{m,\infty}}(g^*_\#\alpha, g^*_\#\beta) \le d_{\mathcal{C}_r^m}(g^*_\#\alpha, g^*_\#\beta) \qquad (4)$$

$$= \sup_{l \in \mathcal{C}_r^m \circ g^*}\Big\{\mathbb{E}_{x \sim \alpha}[l(x)] - \mathbb{E}_{x \sim \beta}[l(x)]\Big\}. \qquad (5)$$

Inequality (4) is based on the observation that there exists $r > 0$ for which $\mathcal{W}_1^{m,\infty} \subset \mathcal{C}_r^m$ [4]. Given any $f \in \mathcal{C}_r^m$ and $g^* \in G_{Lip}$,

$$\|f \circ g^*\|_\infty = \left\{ \sup |f(g^*(y))| : y \in \mathbb{R}^k \right\} = \left\{ \sup |f(x)| : x = g^*(y) \in \mathbb{R}^d, y \in \mathbb{R}^k \right\}$$
$$\leq \left\{ \sup |f(x)| : x \in \mathbb{R}^d \right\} = \|f\|_\infty .$$

Moreover, for $x, y \in \mathbb{R}^k$, $x \neq y$

$$\frac{|D^\alpha f(g^*(x)) - D^\alpha f(g^*(y))|}{|x - y|^{s - \lfloor s \rfloor}} = \frac{|D^\alpha f(g^*(x)) - D^\alpha f(g^*(y))|}{|g^*(x) - g^*(y)|^{s - \lfloor s \rfloor}} \left\{ \frac{|g^*(x) - g^*(y)|}{|x - y|} \right\}^{s - \lfloor s \rfloor}$$
$$\leq \frac{|D^\alpha f(x^*) - D^\alpha f(y^*)|}{|x^* - y^*|^{s - \lfloor s \rfloor}} (L_G)^{s - \lfloor s \rfloor},$$

assuming $x^* \neq y^* \in \mathbb{R}^d$. Here, we choose both the metrics $c, c'$ to be $L^1$ in their respective spaces. This convention conforms to the rest of the discussion as well.

Also, for $1 \leq |s| \leq m$ we have

$$D^s(f \circ g^*)(x) = s! \sum_{1 \leq |i| \leq |s|} \frac{(D^i f)(g^*(x))}{i!} P_{s,i}(g^*; x),$$

where $P_{s,i}(g^*; x)$ is a homogeneous polynomial of degree $|i|$. Schreuder $et\ al.$ [Lemma 7.2 in [5]] show that $\left| D^s(f \circ g^*)(x) \right| < C$, where $C > 0$ is a constant. This implies that there exists $r^* > 0$ for which $f \circ g^* \in \mathcal{C}_{r^*}^m(\mathbb{R}^k)$. As such, we may upper bound (5) by replacing the supremum over $\mathcal{C}_r^m(\mathbb{R}^d) \circ g^*$ by the same over $\mathcal{C}_{r^*}^m(\mathbb{R}^k)$.

Hence, for $\epsilon > 0$

$$d_{\mathcal{W}_1^{m,\infty}}(\hat{\mu}_{n_1}, \phi_\# \hat{\nu}_{n_2}) \leq 2 \inf_{g' \in G_{Lip}} \left\| \phi - g' \right\|_\infty + d_{\mathcal{C}_{r^*}^m}(\nu, \hat{\nu}_{n_2}) + \epsilon.$$

The expected approximation error in the base domain can be put under a deterministic upper bound given by $\mathbb{E}_\nu \left[ d_{\mathcal{C}_{r^*}^m}(\nu, \hat{\nu}_{n_2}) \right] \precsim n_2^{-\frac{m}{k}} + \frac{\log n_2}{\sqrt{n_2}}$ [Lemma 2.8 in [6]]. As such, we get $\mathbb{E}\left[ d_{\mathcal{W}_1^{m,\infty}}(\hat{\mu}_{n_1}, \phi_\# \hat{\nu}_{n_2}) \right] \leq \mathcal{O}\left( n_2^{-\frac{m}{k}} + \frac{\log n_2}{\sqrt{n_2}} \right) + \mathcal{O}(\sqrt{k} L_G B_y W^{-\frac{2}{k}} L^{-\frac{2}{k}}).$ $\qquad\square$

***Proof of Proposition (1).*** Let us denote the VC dimension of $\mathcal{Y}(\mathscr{P}(\mathcal{X}))$ by $v_x < \infty$. This criteria ensures that the target class of distributions are 'learnable'. For example, VC-dim$[\mathcal{Y}(\mathcal{G}_d)] = \mathcal{O}(d^2)$, where $\mathcal{G}_d$ = the class of $d$-dimensional Gaussian distributions [7]. Now, given $g \in G_{Lip}$, for any $n \in \mathbb{N}^+$

$$d_{\mathscr{L}_c^1}(g_\# \hat{\nu}_n, \widehat{(g_\# \nu)}_n) \leq d_{\mathscr{L}_c^1}(g_\# \hat{\nu}_n, g_\# \nu) + d_{\mathscr{L}_c^1}(g_\# \nu, \widehat{(g_\# \nu)}_n)$$
$$\leq L_G\, d_{\mathscr{L}_{c'}^1}(\hat{\nu}_n, \nu) + B_x \left\| g_\# \nu - \widehat{(g_\# \nu)}_n \right\|_{TV}. \qquad (6)$$

Inequality (6) exploits the relation between Wasserstein and TV metrics [Theorem 4 in [8]]. We know there exists constants $\tilde{C}_1, \tilde{C}_2 > 0$ such that

$$\mathbb{P}\left( \left\| g_\# \nu - \widehat{(g_\# \nu)}_n \right\|_{TV} \geq \tilde{C}_1 \sqrt{\frac{v_x}{n}} + t \right) \leq \exp\left( -\tilde{C}_2 n t^2 \right),$$

[Lemma 2 in [9]]. Using this argument along with (2) we obtain

$$\mathbb{P}\left( d_{\mathscr{L}_c^1}(g_\# \hat{\nu}_n, \widehat{(g_\# \nu)}_n) \leq t + \mathcal{O}(n^{-\frac{1}{k}}) + \mathcal{O}(\sqrt{v_x} n^{-\frac{1}{2}}) \right) \geq 1 - 2\exp\left( -C_2 n t^2 \right),$$

where $C_2 = \frac{1}{4} \min \left\{ \frac{2}{(B_y L_G)^2}, \frac{\tilde{C}_2}{B_x^2} \right\} > 0$. As such, the function $g$ is an information preserving map of degree 1, under the 1-Wasserstein metric, with a decaying error of order $\mathcal{O}(n^{-\frac{1}{k \vee 2}})$. $\qquad\square$

**Proof of Lemma (4).** Our characterization of the critics allow $\mathscr{L}_X$ to be $\mathscr{L}_c^1$ or $\mathcal{W}_1^{m,\infty}$. Under this setup, for any backward translator $G$

$$d_{\mathscr{L}_X}(\hat{\mu}_{n_1}, G_\#\hat{\nu}_{n_2}) \leq d_{\mathscr{L}_X}(\hat{\mu}_{n_1}, \widehat{(G_\#\nu)}_{n_2}) + d_{\mathscr{L}_X}(\widehat{(G_\#\nu)}_{n_2}, G_\#\hat{\nu}_{n_2}) \tag{7}$$

$$\leq B_x \left\| \hat{\mu}_{n_1} - \widehat{(G_\#\nu)}_{n_2} \right\|_{TV} + \mathcal{E}_3$$

$$\leq B_x \left\| \hat{\mu}_{n_1} - \Gamma_{n_1} \right\|_{TV} + \Lambda_{(n_1, n_2)} + \mathcal{E}_3,$$

where $\Gamma_{n_1} = \mathrm{argmin}_{\tau \in \mathscr{P}(\mathcal{X})} \|\tau - \hat{\mu}_{n_1}\|_{TV}$. It is often called the *Empirical Yatracos Minimizer* [10]. Observe that, $\|\hat{\mu}_{n_1} - \Gamma_{n_1}\|_{TV} \leq \|\hat{\mu}_{n_1} - \mu\|_{TV}$. Now, in case the OT map $T$ exists such that $T_\#\nu = \mu$, we get $\|\hat{\mu}_{n_1} - \Gamma_{n_1}\|_{TV} \leq \mathcal{E}_1$. $\square$

**Remark.** The information loss (in the right-hand side of (7)) can be taken care of by deploying an IPT as the translator. As such, it is the term $d_{\mathscr{L}_X}(\hat{\mu}_{n_1}, \widehat{(G_\#\nu)}_{n_2})$ that mainly contributes to the upper bound. We had built the empirical distribution $\hat{\mu}_{n_1}$ based on $\{X_i\}_{i=1}^{n_1} \overset{i.i.d.}{\sim} \mu$. Similarly, let $\widehat{(G_\#\nu)}_{n_2}$ be based on $\{Y_i\}_{i=1}^{n_2} \overset{i.i.d.}{\sim} G_\#\nu$. We may write

$$d_{\mathscr{L}_X}(\hat{\mu}_{n_1}, \widehat{(G_\#\nu)}_{n_2}) = \sup_{f \in \mathscr{L}_X} \left| \sum_{i=1}^{N} W_i f(Z_i) \right|, \tag{8}$$

where $N = n_1 + n_2$; $W_i = \frac{1}{n_1}$ when $Z_i = X_i$, $i = 1, ..., n_1$ and $W_{n_1+j} = -\frac{1}{n_2}$ when $Z_{n_1+j} = Y_j$, $j = 1, ..., n_2$. Under this framework, the solution to (8) can be achieved by solving a linear program, given that $\mathscr{L}_X \equiv \mathscr{L}_c^1$ [Theorem 2.1 in [11]]. This provides a pathway to get hold of the realized approximation error, making the upper bound deterministic.

**Proof of Lemma (5).** Given translator maps $G \in \mathscr{F}(\mathcal{Y}, \mathscr{P}(\mathcal{X}))$ and $F \in \mathscr{F}(\mathcal{X}, \mathscr{P}(\mathcal{Y}))$, the cyclic loss in the space $\mathcal{X}$ can be broken down as the following:

$$\left\| \mu - (G \circ F)_\#\mu \right\|_1 \leq \left\| \mu - G_\#\nu \right\|_1 + \left\| G_\#\nu - (G \circ F)_\#\mu \right\|_1,$$

where

$$\left\| G_\#\nu - (G \circ F)_\#\mu \right\|_1 = \left\| G_\#\nu - G_\#(F_\#\mu) \right\|_1 = 2 \sup_{\omega \subseteq \sigma(\mathcal{X})} \left| G_\#\nu(\omega) - G_\#(F_\#\mu)(\omega) \right|$$

$$= 2 \sup_{\omega \subseteq \sigma(\mathcal{X})} \left| \nu(G^{-1}(\omega)) - F_\#\mu(G^{-1}(\omega)) \right|$$

$$\leq 2 \sup_{\omega' \subseteq \sigma(\mathcal{Y})} \left| \nu(\omega') - F_\#\mu(\omega') \right| = \left\| \nu - F_\#\mu \right\|_1.$$

The inequality holds by taking supremum over all measurable sets belonging to the Borel $\sigma$-algebra on $\mathcal{Y}$ instead of the particular path directed by $G^{-1}$. As such

$$\left\| \mu - (G \circ F)_\#\mu \right\|_1 \leq \left\| \mu - G_\#\nu \right\|_1 + \left\| \nu - F_\#\mu \right\|_1.$$

Similarly, $\left\| \nu - (F \circ G)_\#\nu \right\|_1 \leq \left\| \nu - F_\#\mu \right\|_1 + \left\| \mu - G_\#\nu \right\|_1$. Hence the proof. $\square$

**Proof of Theorem (3).** Given a measurable function $f : \mathbb{R}^d \to \mathbb{R}$, let us define its *convolution* with the kernel $K : \mathbb{R}^d \times \mathbb{R}^d \to \mathbb{R}$ as the following:

$$K_h(f) = \int_{\mathbb{R}^d} K_h(., y) f(y) dy = \frac{1}{h^d} \int_{\mathbb{R}^d} K(\frac{.}{h}, \frac{y}{h}) f(y) dy,$$

where $\frac{y}{h} = (\frac{y_1}{h}, ..., \frac{y_d}{h})'$, $h > 0$. We begin by taking $K$ to be regularly invariant. Now,

$$\left\| p_\mu - p_{\phi_\#\nu} \right\|_1 \leq \left\| p_\mu - K_h(p_\mu) \right\|_1 + \left\| K_h(p_\mu) - K_h(p_{\phi_\#\nu}) \right\|_1 + \left\| K_h(p_{\phi_\#\nu}) - p_{\phi_\#\nu} \right\|_1$$

$$\leq J \left\| p_\mu - K_h(p_\mu) \right\|_p + \left\| K_h(p_\mu) - K_h(p_{\phi_\#\nu}) \right\|_1 + J \left\| K_h(p_{\phi_\#\nu}) - p_{\phi_\#\nu} \right\|_{p'},$$

$$\tag{9}$$

where $J > 0$. The existence of such a constant, and hence the inequality (9), is ensured by the fact $\|f\|_1 \leq J\|f\|_p, p \geq 1$ since we have $\lambda(\Omega_x) < \infty$. Also, there exists a constant $l$ depending upon $m_x$ and $K$, such that $\left\|K_h(p_\mu) - p_\mu\right\|_p \leq l\left\|D^{m_x}p_\mu\right\|_p h^{m_x}$ [Proposition 4.3.33 in [12]]. As such, we get hold of a constant $J^* = Jl$ for which

$$\left\|p_\mu - p_{\phi_\#\nu}\right\|_1 \leq J^*\left\{\left\|D^{m_x}p_\mu\right\|_p + \left\|D^{m_x}p_{\phi_\#\nu}\right\|_{p'}\right\}h^{m_x} + \left\|K_h(p_\mu) - K_h(p_{\phi_\#\nu})\right\|_1$$

(by Assumption 2). Observe that,

$$K_h(p_\mu)(x) - K_h(p_{\phi_\#\nu})(x) = \frac{1}{h^d}\int\left\{K(\frac{x}{h}, \frac{y}{h}) - K(\frac{x}{h}, \frac{z}{h})\right\}d\kappa(y, z),$$

where $\kappa$ is a coupling between $\mu$ and $\phi_\#\nu$. Hence,

$$\left\|K_h(p_\mu) - K_h(p_{\phi_\#\nu})\right\|_1 \leq \int\left\{\frac{1}{h^d}\int\left|K(\frac{x}{h}, \frac{y}{h}) - K(\frac{x}{h}, \frac{z}{h})\right|dx\right\}d\kappa(y, z) \tag{10}$$

$$= \int\left\{\frac{\int\left|K(x', \frac{y}{h}) - K(x', \frac{z}{h})\right|dx'}{|y - z|}\right\}|y - z|\,d\kappa(y, z)$$

$$\leq \frac{M^*}{h}\int|y - z|\,d\kappa(y, z), \tag{11}$$

where $M^*$ is a positive constant. The step (10) is due to Jensen's inequality, whereas (11) exploits the invariance of $K$. Since the inequality holds for all possible measure couples $\kappa$, we conclude

$$\left\|K_h(p_\mu) - K_h(p_{\phi_\#\nu})\right\|_1 \leq \frac{M^*}{h}W_c^1(\mu, \phi_\#\nu),$$

given that $c \equiv L^1$. A similar inference can be drawn for a general class of metrics $c$ by altering the specification of the same in the definition of invariance. Now, choose

$$h = \left\{\frac{W_c^1(\mu, \phi_\#\nu)}{\left\|D^{m_x}p_\mu\right\|_p + \left\|D^{m_x}p_{\phi_\#\nu}\right\|_{p'}}\right\}^{\frac{1}{m_x+1}}.$$

Finally, we obtain

$$\left\|p_\mu - p_{\phi_\#\nu}\right\|_1 \leq M\left[\left\|D^{m_x}p_\mu\right\|_p + \left\|D^{m_x}p_{\phi_\#\nu}\right\|_{p'}\right]^{\frac{1}{m_x+1}}\left[W_c^1(\mu, \phi_\#\nu)\right]^{\frac{m_x}{m_x+1}},$$

where $M = 2(J^* \vee M^*)$. $\qquad\square$

***Proof of Proposition (2).*** Using Lemma (5),

$$\mathcal{L}_{cyc}(\hat{\mu}_{n_1}, \hat{\nu}_{n_2}, F, G) = \left\|\hat{\mu}_{n_1} - (G \circ F)_\#\hat{\mu}_{n_1}\right\|_1 + \left\|\hat{\nu}_{n_2} - (F \circ G)_\#\hat{\nu}_{n_2}\right\|_1$$

$$\leq 4\left\{\left\|\hat{\mu}_{n_1} - G_\#\hat{\nu}_{n_2}\right\|_{TV} + \left\|\hat{\nu}_{n_2} - F_\#\hat{\mu}_{n_1}\right\|_{TV}\right\}.$$

Now, a similar decomposition of the translation errors under the TV metric, as in the proof of Lemma (4), results in the following:

$$\left\|\hat{\mu}_{n_1} - G_\#\hat{\nu}_{n_2}\right\|_{TV} \leq \left\|\hat{\mu}_{n_1} - \Gamma_{n_1}\right\|_{TV} + \left\|\Gamma_{n_1} - \widehat{(G_\#\nu)}_{n_2}\right\|_{TV} + \left\|\widehat{(G_\#\nu)}_{n_2} - G_\#\hat{\nu}_{n_2}\right\|_{TV}$$

$$\leq \left\|\hat{\mu}_{n_1} - \mu\right\|_{TV} + \frac{\Lambda_{(n_1, n_2)}}{B_x} + \left\|\widehat{(G_\#\nu)}_{n_2} - G_\#\hat{\nu}_{n_2}\right\|_{TV}.$$

Similarly, given that $\Gamma'_{n_2} = \text{argmin}_{\tau \in \mathscr{P}(\mathcal{Y})}\|\tau - \hat{\nu}_{n_2}\|_{TV}$

$$\left\|\hat{\nu}_{n_2} - F_\#\hat{\mu}_{n_1}\right\|_{TV} \leq \left\|\hat{\nu}_{n_2} - \nu\right\|_{TV} + \frac{\Lambda'_{(n_1, n_2)}}{B_y} + \left\|\widehat{(F_\#\mu)}_{n_1} - F_\#\hat{\mu}_{n_1}\right\|_{TV}.$$

$\qquad\square$

**Proof of Theorem (4).** Let $\phi \in \Phi(W, L)_k^d$, as specified in Theorem (1). Also, let $\psi \in \Phi(W', L')_d^k$ be a forward translator that achieves consistency. Observe that

$$\hat{\mathcal{L}}_{cyc}(\tilde{\mu}_{n_1}, \tilde{\nu}_{n_2}, \psi, \phi) \leq \|\tilde{\mu}_{n_1} - \mu\|_1 + \|\tilde{\nu}_{n_2} - \nu\|_1 + \mathcal{L}_{cyc}(\mu, \nu, \psi, \phi)$$

$$\leq \|\tilde{\mu}_{n_1} - \mu\|_1 + \|\tilde{\nu}_{n_2} - \nu\|_1 + 2\left\{\|\mu - \phi_{\#}\nu\|_1 + \|\nu - \psi_{\#}\mu\|_1\right\}. \quad (12)$$

For $1 \leq p, q < \infty$, we know that

$$\mathbb{E}\left[\|\hat{p}_{\mu,n_1} - p_\mu\|_p\right] \precsim n_1^{-\frac{m_x}{2m_x+d}},$$

[Theorem 6.1 in [13]]. Similarly, for the estimation error in $\mathcal{Y}$, $\mathbb{E}\left[\|\hat{p}_{\nu,n_2} - p_\nu\|_q\right] \precsim n_2^{-\frac{m_y}{2m_y+k}}$. Moreover, Theorem (3) implies that

$$\left\{\|p_\mu - p_{\phi_{\#}\nu}\|_1\right\}^{\frac{m_x+1}{m_x}} \leq R\, d_{\mathscr{L}_c^1}(\mu, \phi_{\#}\nu) \leq R\left\{d_{\mathscr{L}_c^1}(\mu, \hat{\mu}_{n_1}) + d_{\mathscr{L}_c^1}(\hat{\mu}_{n_1}, \phi_{\#}\nu)\right\}, \quad (13)$$

where $R = M^{\frac{m_x+1}{m_x}}\left[\|D^{m_x}p_\mu\|_p + \|D^{m_x}p_{\phi_{\#}\nu}\|_{p'}\right]^{\frac{1}{m_x}}$, and $\hat{\mu}_{n_1}$ is an usual empirical measure corresponding to $\mu$. The term $d_{\mathscr{L}_c^1}(\hat{\mu}_{n_1}, \phi_{\#}\nu)$ can be made arbitrarily small due to the construction of $\phi$ [Lemma (1)]. Also, we have already seen that $\mathbb{E}\left[d_{\mathscr{L}_c^1}(\mu, \hat{\mu}_{n_1})\right] \precsim n_1^{-\frac{1}{d}}$.

As such,

$$\mathbb{E}\left[\|\tilde{\mu}_{n_1} - \mu\|_1 + 2\|\mu - \phi_{\#}\nu\|_1\right] \leq \mathcal{O}\left(n_1^{-\frac{m_x}{(d\vee 2)m_x+d}}\right),$$

by applying Jensen's inequality to (13). This bound, together with a similar result corresponding to its forward counterpart, will imply

$$\mathbb{E}\left[\hat{\mathcal{L}}_{cyc}(\tilde{\mu}_{n_1}, \tilde{\nu}_{n_2}, \psi, \phi)\right] \precsim \max\left\{n_1^{-\frac{m_x}{(d\vee 2)m_x+d}}, n_2^{-\frac{m_y}{(k\vee 2)m_y+k}}\right\}.$$

$\square$

**Proof of Corollary (2).** We point out that, $K(x, y)$ can be taken in particular as $\tilde{K}(|x - y|)$, where $\tilde{K} : \mathbb{R}^d \to \mathbb{R}$ identically follows the traits of $K$. Under such a kernel function,

$$\left\|\mathbb{E}[\hat{p}_{\mu,n_1}] - p_\mu\right\|_1 \leq l^* h^{m_x},$$

for some constant $l^* > 0$ [12]. Now, given an $\epsilon \leq \frac{2}{3}$, concentration inequalities on kernel density estimates tell us: there exists constants $E_1, E_2 > 0$ such that

$$\mathbb{P}\left(\left\|\hat{p}_{\mu,n_1} - \mathbb{E}[\hat{p}_{\mu,n_1}]\right\|_\infty > \epsilon\right) \leq E_1\left(\frac{\sqrt{d}B_x}{h^{d+1}\epsilon}\right)^d \exp\left(-E_2 n_1 \epsilon^2 h^d\right).$$

The exact value of $E_2 = \frac{3}{28\tilde{K}(0)}$ can be obtained based on the convention that $\tilde{K}(.)$ achieves its modal value at $0$. Such a centering can always be done. Hence,

$$\mathbb{P}\left(\left\|\hat{p}_{\mu,n_1} - p_\mu\right\|_1 > \epsilon + l^* h^{m_x}\right) \leq E_1\left(\frac{\sqrt{d}B_x}{h^{d+1}\epsilon}\right)^d \exp\left(-E_2 n_1 \epsilon^2 h^d\right). \quad (14)$$

By applying Borel-Cantelli lemma one can show that $\|\hat{p}_{\mu,n_1} - p_\mu\|_1 \longrightarrow 0$ almost surely, under suitable choice of $h \equiv h(n_1, m_x, d)$. (14) inspires a similar concentration for the estimate $\hat{p}_{\nu,n_2}$ around $p_\nu$, under $L^1$. As such, by taking the corresponding bandwidth $h' \equiv h'(n_2, m_y, k)$, it can also be said that $\|\hat{p}_{\nu,n_2} - p_\nu\|_1 \longrightarrow 0$ almost surely. To unify the two processes, one may assess the convergence based on $n = \min\{n_1, n_2\}$. Putting these results back in (12), along with (13), we conclude

$$\hat{\mathcal{L}}_{cyc}(\tilde{\mu}_{n_1}, \tilde{\nu}_{n_2}, \psi, \phi) \longrightarrow 0, \text{ almost surely.}$$

In other words, $(\phi \circ \psi)_{\#}\tilde{\mu}_{n_1} \to \mu$ and $(\psi \circ \phi)_{\#}\tilde{\nu}_{n_2} \to \nu$, both in total variation. $\square$

## Identity loss

Let us first rewrite the identity loss in terms of the underlying measures. Based on the notations in our framework,

$$\mathcal{L}_{id}(\mu, \nu, F, G) = \left\| \mu - F_{\#}\mu \right\|_1 + \left\| \nu - G_{\#}\nu \right\|_1.$$

Observe that the distributions must be equivariate to conform to this loss. Moreover,

$$\left\| \mu - \nu \right\|_1 - \left\| F_{\#}\mu - \nu \right\|_1 \le \left\| \mu - F_{\#}\mu \right\|_1. \tag{15}$$

If the forward translated law $F_{\#}\mu$ is Sobolev-smooth of order $m_y$ (Assumption 2), Theorem (3) asserts the existence of a constant $R' > 0$ such that $\left\| p_\nu - p_{F_{\#}\mu} \right\|_1 \le R' \left[ d_{\mathscr{L}_c^1}(\nu, F_{\#}\mu) \right]^{\frac{m_y}{m_y+1}}$. In case $F$ is also translation consistent, the second term on the left-hand side of (15) vanishes. A similar conclusion can be drawn for the quantity $\left\| \nu - G_{\#}\nu \right\|_1$ as well. As such, the cumulative identity loss from both domains cannot be minimized beyond the intrinsic discrepancy between the input distributions.