# OpenReview forum: "On Translation and Reconstruction Guarantees of the Cycle-Consistent Generative Adversarial Networks"
_NeurIPS.cc/2022/Conference — NeurIPS 2022 Accept_

### Official Review · Reviewer_JEYa · 2022-07-10

**Rating:** 6
**Confidence:** 1
**Soundness:** 3 good
**Presentation:** 3 good
**Contribution:** 3 good

**Summary:**

I do not think my expertise matches this paper so my summary of the paper and its contributions may not be accurate. This paper provides a series of math explanations and proof for the statistical translation and regeneration guarantees of cycle-consistent networks. The contribution of the paper is establishing an analytical way for not only cycle-consistent networks but also other deep generative models.

**Questions:**

N/A.

**Limitations:**

N/A.

**Strengths And Weaknesses:**

Strengths:
There are rare papers that provide the translation of the deep generative models based on rigorous mathematics. Therefore, I do think this paper has high originality and may have a good impact on the readers to conduct further research in the future.

Weaknesses:
Although math is a beautiful way to elaborate an algorithm, sometimes telling a story by images has more affinity to the readers and may help readers to know the whole picture faster.

---

> ### Author Response · Authors · 2022-07-30
> **Response to Reviewer JEYa**
>
> We thank the reviewer for the encouraging and constructive comments. We also appreciate the reviewer's acknowledgement of our originality.
>
> We will certainly include a pictorial representation of the underlying density estimation problems (simultaneous translations and reconstructions), pointing out the roles of each component involved.

---

> > ### Author Response · Authors · 2022-08-02
> > **Further Response to Reviewer JEYa**
> >
> > In the supplementary material, we have added diagrams highlighting the simultaneous translations and regeneration in the space $\\mathcal{Y}$. This is to maintain space economy in the main paper, and we will integrate the illustrations with our discussions seamlessly in the final version.

---

### Official Review · Reviewer_rk2p · 2022-07-10

**Rating:** 6
**Confidence:** 3
**Soundness:** 3 good
**Presentation:** 3 good
**Contribution:** 3 good

**Summary:**

In this paper, the authors propose a novel statistic framework to analyze the consistency of Cycle-GAN type neural networks, where translator and inverse-translator between two domains in the images spaces are trained. In particular, given ReLU DNN translators, the authors prove that they are information preserving based on delicate functional analysis tools. Moreover, the paper provides theoretical attests to previous empirical studies on the equivalence of L1 norm and 1-W distance in cyclic loss. It is also proved that using translation consistent translators, a cyclic-consistent network can be built.

**Questions:**

- in line 374, would you elaborate more on the self-attacking behavior?


**Limitations:**

The limitations of the paper are discussed in the last section.

**Strengths And Weaknesses:**

The strengths of the paper include rigorous statistical analysis to studying the cyclic-consistency of cycle-GAN type networks: sharp non-asymptotic bounds on the translation error under mild conditions are rigorously proven. Also, for the first time, a deterministic bound on cumulative reconstruction error is given. This sheds new light on where the ill-posedness of cycle-GAN type networks stems from. The results are clearly stated and the motivations are explained well. Also,  commonly seen cycle-GAN type networks are reviewed and compared under the proposed framework, showing the great value in the application of the proposed analyzing framework.

It might be more illustrative if numerical experiment results are given to support the theoretical results. But since the paper is fruitful theoretically, it is only a minor drawback.

---

> ### Author Response · Authors · 2022-07-30
> **Response to Reviewer rk2p**
>
> We thank the reviewer for the inspiring and pertinent comments.
>
> $\mathbf{Q1.}$ We thank the reviewer for this suggestion. Cyclic image-to-image translation models, where the desired map is many-to-one (e.g. photos to semantic labels), often suffer from this issue. As discussed with much rigour in [4], this phenomenon can be interpreted as the translators' tendency to 'hide information inside tiny perturbations of the translated image'. As a result, translations are often erroneous, although reconstructions turn out to be near-perfect. Statistical explanations addressing this problem remain largely absent. For a detailed elucidation based on empirical evidence, we refer the reader to [4]. They also prescribe two defense mechanisms against self-attack (1. Adversarial training with noise, 2. Using Guess discriminators). We will include the related discussions in greater detail in the final version of our paper.
>
> [4] Dina Bashkirova, Ben Usman, and Kate Saenko, Adversarial Self-Defense for Cycle-Consistent GANs, Proceedings of the 33rd International Conference on Neural Information Processing Systems, 2019.

---

### Official Review · Reviewer_KfH7 · 2022-07-12

**Rating:** 5
**Confidence:** 2
**Soundness:** 3 good
**Presentation:** 2 fair
**Contribution:** 3 good

**Summary:**

In this paper the authors analyze the properties of image to image translation networks that are based on cycle consistency.
In particular, the authors show that when the translation networks are based on ReLU activations they behave as Information preserving transformations (IPTs). Next the authors also show that 1-Wasserstein distance and L1 cyclic distance are equivalent for Sobolev-smooth data and analyze the effects of ill-posedness on the regeneration.

**Questions:**

Additional discussion about other architectural choices for the translation networks should be helpful.
Moreover, description of how these assumptions correspond to existing image-to-image translation methods should be very useful.

**Limitations:**

The relevant assumptions about the network architectures and the data are stated clearly by the authors

**Strengths And Weaknesses:**

The setting is rigorously formulated and all the claims made in the manuscript appear to be shown in ample detail.

---

> ### Author Response · Authors · 2022-07-30
> **Response to Reviewer KfH7**
>
> We thank the reviewer for the positive review and constructive suggestions thereafter.
>
> $\mathbf{Q1.}$ Our motivation behind selecting ReLU-based translator networks lies in their capability of approximating functions of various kinds, and hence possible transport (optimal or sub-optimal) maps. One particular property that turns out to be crucial is their near-perfect behaviour as an Information Preserving Transform (IPT). Neural networks based on tanh [1], sigmoid [2], and GroupSort [3] activations have also been shown to approximate Lipschitz functions with high precision. However, the information preserving property needs to be checked for all of them. We will definitely add this point in the revised version.
>
> [1] Tim De Ryck, Samuel Lanthaler, and Siddhartha Mishra, On the approximation of functions by tanh neural networks, Neural Networks, Volume 143, 2021.
>
> [2] Sophie Langer, Approximating smooth functions by deep neural networks with sigmoid activation function, arXiv:2010.04596, 2020.
>
> [3] Ugo Tanielian, Maxime Sangnier, and Gerard Biau, Approximating Lipschitz continuous functions with GroupSort neural networks, arXiv:2006.05254, 2020.
>
>
>
> $\mathbf{Q2.}$ In the paper, we make assumptions about the smoothness of the data distributions and the compactness of their supports. Such restrictions are mild and also widely applied in related literature. Feature extracted image data can be viewed as real vectors with bounded entries in each dimension. As such, the bounded nature of the support is meaningful. However, similar analyses can be carried out without the assumption in place. In case of unbounded support, if the data distributions have sharp decaying tails (e.g. sub-exponential), comparable convergence results can be achieved. Being fairly general, we believe our assumptions can be called upon while analysing any data translation method. We will make sure to point this out in the article. We also emphasize the fact that the absence of prior theoretical studies on image-to-image translation methods of this spirit restricts us from providing a comparative discussion on them.
>
>
> $\\bullet$ Given the positive comments from your side and taking into account the other reviews and our responses, we fervently hope you will consider revising the score for this paper. Also, we would be more than happy to address any further questions you have since we are approaching the end of the discussion period.

---

### Official Review · Reviewer_z1Ah · 2022-07-13

**Rating:** 7
**Confidence:** 3
**Soundness:** 4 excellent
**Presentation:** 3 good
**Contribution:** 3 good

**Summary:**

In this paper, the authors provided their theoretical understanding on the translation and reconstruction guarantees of the cycle-consistent GANs. They showed that the translators based on deep ReLU networks can prevent the input information from dissipating during generation cycles, which means we can lower bound the gap. On the other hand, there are positive results. The authors proved that the same translators not only achieve zero generation loss asymptotically but the generated distribution converges to the target distribution almost surely. The authors also showed the equivalence between L1 norm and 1-Wasserstein distance in the cyclic loss.

**Questions:**

Is it possible to take the optimization trajectory or process into consideration, since we know that GAN is notorious for its hardness of training and instability of training.

**Limitations:**

I think there is no potential negative societal impact of this work since it is completely theoretical.

**Strengths And Weaknesses:**

This paper is very solid in the theory. The writing quality of this paper is high. And most importantly, the problem analyzed by the authors is very fundamental and important in the field of generative models.

Although the literature review of this paper does not cover all the theoretical papers on GANs, it is not a big deal. The results are original and significant.

---

> ### Author Response · Authors · 2022-07-30
> **Response to Reviewer z1Ah**
>
> We thank the reviewer for the positive assessment and acknowledgement of our contributions.
>
> As the reviewer has rightly pointed out, the literature review we provided in the paper is not exhaustive. Very recently, numerous theoretical studies have been reported on GAN and its variants, highlighting various aspects. We feel it to be beyond the capacity of this paper to acknowledge most, if not all, of their contributions. The brevity of the article binds us to be concise. Our brief survey is merely an attempt to honour the papers that inspire us in our pursuit without losing context to the reader. In the final version, however, we will make the survey more exhaustive, self-contained, and still aligned with the current study.
>
> $\\mathbf{Q1.}$ Thank you for this valuable comment. We follow a non-parametric approach in depicting translations as density estimation tasks. In other words, we do not recognize the set of candidate distributions to be exactly characterized by an underlying parameter space (say, $\\Theta$). The translated law (e.g. $G_{\\#}{\hat{\\nu}}\_n$) tries to close its gap from the target distribution ($\\mu$) by approaching the minimum distance estimate. On the other hand, training can be viewed as the process of finding out a suitable parameter value (say, $\hat{\\theta} \\in \\Theta$) such that the corresponding density estimate (say, $p_{\\hat{\\theta}}$) optimizes the loss. This is crucial since, during training, the parameters of the translator networks are responsible for shaping up this parameter space, and hence the optimum $\hat{\theta}$. In this regime, the optimization trajectory can accordingly be viewed as the stochastic propagation of $\hat{\theta}$ (as a function of $n$: the input sample size, and $t$: iterations or time) towards a stable value. We do not feel a non-parametric study can serve this problem justice. Instead, we must recognize this line as a potential future work.
>
> We also point out that incorporating the training process in a parametric regime brings along questions regarding domain adaptation. That is, the performance of $p_{\hat{\theta}}$ during testing needs to be considered as well. We will add some relevant discussion in the final version of our paper.

---

### Meta-Review · Area_Chair_KGva · 2022-08-24

**Recommendation:** Accept
**Confidence:** Certain

**Metareview:**

This paper analyzes the consistency of cycle-consistent GANs. The authors provide theoretical insights in how the information is preserved by ReLU DNN translators via functional analysis. They additionally show the equivalence of L1 and 1-W distance under the cyclic loss context. The highlight of this paper is the rigorous statistical analysis and fundamental theoretical insights in generative models. A minor point is that the reviewers suggest that more illustrative and intuitive presentation of the results will smooth the reading. In general it's a interesting paper and the AC recommends acceptance.

**Award:**

No

---

### Decision · Program_Chairs · 2022-09-14

Accept